# Loop-Mediated Isothermal Amplification Allows Rapid, Simple and Accurate Molecular Diagnosis of Human Cutaneous and Visceral Leishmaniasis Caused by *Leishmania infantum* When Compared to PCR

**DOI:** 10.3390/microorganisms9030610

**Published:** 2021-03-16

**Authors:** Ana Victoria Ibarra-Meneses, Carmen Chicharro, Carmen Sánchez, Emilia García, Sheila Ortega, Joseph Mathu Ndung’u, Javier Moreno, Israel Cruz, Eugenia Carrillo

**Affiliations:** 1WHO Collaborating Centre for Leishmaniasis, National Center for Microbiology, Instituto de Salud Carlos III, 28220 Majadahonda, Spain; ana.ibarrameneses@gmail.com (A.V.I.-M.); cchichar@isciii.es (C.C.); csanchezh@isciii.es (C.S.); egdiez@isciii.es (E.G.); sheilaortega@isciii.es (S.O.); javier.moreno@isciii.es (J.M.); ecarrillo@isciii.es (E.C.); 2Foundation for Innovative New Diagnostics, 1202 Geneva, Switzerland; Joseph.Ndungu@finddx.org; 3National School of Public Health, Instituto de Salud Carlos III, 28029 Madrid, Spain

**Keywords:** loop-mediated isothermal amplification, LAMP, Loopamp, diagnosis, cutaneous leishmaniasis, visceral leishmaniasis, *Leishmania infantum*, PCR

## Abstract

Loop-mediated isothermal amplification allows the rapid, sensitive and specific amplification of DNA without complex and expensive equipment. We compared the diagnostic performance of Loopamp™ *Leishmania* Detection Kit (Eiken Chemical Co., Ltd., Tokyo, Japan) with conventional and real-time polymerase chain reaction (PCR) for human cutaneous and visceral leishmaniasis caused by *L. infantum*. A total of 230 DNA samples from cutaneous (CL) and visceral (VL) leishmaniasis cases and controls from Spain, characterized by *Leishmania* nested PCR (LnPCR) were tested by: (i) the Loopamp™ *Leishmania* Detection Kit (Loopamp), run on Genie III real-time fluorimeter (OptiGene, UK); and (ii) real-time quantitative PCR (qPCR). The Loopamp test returned 98.8% (95% confidence interval—CI: 96.0–100.00) sensitivity and specificity of 97.7% (95% CI: 92.2–100) on VL samples, and 100% (95% CI: 99.1–100) sensitivity and 100.0% (95% CI: 98.8–100.0) specificity on CL samples. The Loopamp time-to-positivity (*Tp*) obtained by real-time fluorimetry showed excellent concordance (C = 97.91%) and strong correlation (r = 0.799) with qPCR’s cycle threshold (*Ct*). The performance of Loopamp is comparable to that of LnPCR and qPCR in the diagnosis of cutaneous and visceral leishmaniasis due to *L. infantum*. The excellent correlation between the *Tp* and *Ct* should be further investigated to determine the accuracy of Loopamp to quantify parasite load in tissues.

## 1. Introduction

Leishmaniasis is a vector-borne parasitic disease caused by more than 20 species of *Leishmania*. More than 90 sand fly species can transmit leishmaniasis when biting for a blood meal. Clinical presentation in humans is varied but two main forms exist, cutaneous and visceral leishmaniasis. Cutaneous leishmaniasis (CL) is the most common, and about 0.6 to 1 million new cases are reported annually, these are caused by many different *Leishmania* species. Visceral leishmaniasis (VL) can be lethal without early treatment, is caused by the species *Leishmania donovani* and *L. infantum*; between 50,000 and 90,000 new cases occur annually, and being lethal without treatment makes it one of the deadliest parasitic disease [1].

Confirmatory diagnosis of CL and VL requires visualization under the microscope of *Leishmania* amastigotes in Giemsa stained smears prepared from tissue samples, from the lesions in CL, and from bone marrow or spleen aspirates in VL [2]. Microscopy has some limitations in terms of sensitivity and required skills to achieve an accurate result, and as such, even in high burden countries, the number of cases with a confirmatory diagnosis can be as low as 5% for CL and 60% for VL. Parasitological confirmation is key in the diagnosis of CL, and while VL can be diagnosed using serological tests, chiefly in immunocompetent patients, parasitological confirmation may be required as part of the diagnostic algorithm, as well as for test-of-cure or as inclusion criteria in drug clinical trials [3,4,5].

Molecular diagnosis based on polymerase chain reaction (PCR), has been widely used in the diagnosis of CL and VL and presents a number of advantages, such as increased sensitivity and the use of less invasive samples, showing high diagnostic accuracy for VL even using peripheral blood, a sample that usually yields lower sensitivity. Unfortunately, its cost and expertise required alongside the need for a cold chain and specialized laboratories, has prevented its implementation in most endemic settings [6,7].

A molecular diagnostic test that presents less limitations for implementation is the loop-mediated isothermal amplification (LAMP), a robust method that allows for the rapid, sensitive, and specific amplification of target DNA using less sophisticated equipment [8]. A LAMP test developed by Eiken Chemical Co., Ltd., Tokyo, Japan (Loopamp™ *Leishmania* Detection Kit), which is a dry reagent-based test, simple to operate and that does not require a cold chain, has been shown to be useful in the diagnosis of CL and VL in different endemic areas [9,10,11,12]. In a previous study, we demonstrated the analytical performance of Loopamp™ *Leishmania* detection kit [13]. In the present work, we go a step further and assess its clinical performance using a panel of well-characterized clinical samples from CL and VL suspected cases diagnosed at the World Health Organization Collaborating Centre for Leishmaniasis (WHO-CCL) in Spain.

## 2. Materials and Methods

### 2.1. Study Samples

We used a convenience assembly of DNA samples from 230 cases of suspected CL and VL (Table 1) that had a previous LnPCR result. The samples, collected between 2016 and 2018, belong to the collection of the WHO-CCL, at the National Centre for Microbiology, Instituto de Salud Carlos III, Madrid, Spain. The WHO-CCL is a reference laboratory for leishmaniasis and supports hospitals across the country in the diagnosis of leishmaniasis.

### 2.2. Sample Preparation

The stored DNA had been prepared as per the standard protocol at WHO-CCL, briefly: starting from either:Samples from VL suspects: 200 μL of ethylene diamine tetra-acetic acid (EDTA)-treated peripheral blood, bone marrow.Samples from CL suspects: 200 μL skin biopsy (usually 5 mm diameter punch) macerated in NET10 buffer (10 mM NaCl, 10 mM EDTA, 10 mM Tris-HCl, pH 8.0).

Samples were processed using the QIAamp DNA Mini Kit (QIAGEN, Hilden, Germany) following instructions provided by the manufacturer. The DNA was eluted in 200 μL PCR grade water and stored at −20 °C until analysis.

Sample testing workflow (STARD diagram) is described in Appendix A. An additional file shows the STARD checklist (Appendix A). Operators of the different molecular tests were blinded to the previous results of the samples included in the panel.

### 2.3. Reference Test: Leishmania-Nested PCR (LnPCR)

The *Leishmania* nested PCR targets the small subunit rRNA (*SSU rRNA*) gene, a region that is amplified with the sequential use of two pairs of primers, the second pair is the same as in the qPCR described above. Then, 10 microliters DNA were used to run the LnPCR as described elsewhere on an Applied Biosystems 2700 (Foster City, CA, USA) conventional thermocycler, and the final products were resolved by agarose gel electrophoresis [14]. All samples had been previously tested by this method (Table 1), but testing was repeated for the purpose of this study to generate results at the same time as index tests (below) and to confirm original results.

### 2.4. Index Test-1: Loop-Mediated Isothermal Amplification (Loopamp)

Loopamp reactions were performed using the Loopamp™ *Leishmania* Detection Kit (Eiken Chemical Co., Ltd., Tokyo, Japan), which targets both *SSU rRNA* gene and kinetoplast DNA (kDNA) minicircles of *Leishmania*. In total, three microliters DNA were used to run the Loopamp reactions following the instructions provided with the kit and as described elsewhere [13]. The Loopamp reactions were run on a Genie III^®^ real-time fluorimeter (OptiGene Ltd., Horsham, UK). For positive results, the time-to-positivity (*Tp*) was recorded. Additionally, results were also assessed by visual inspection with the naked eye under blue LED light illumination; positive samples emit a green fluorescent light, while negative samples do not emit any light.

### 2.5. Index Test-2: Real-Time Quantitative PCR (qPCR)

This qPCR targets the SSUrRNA gene of Leishmania. Testing was conducted, as described elsewhere, using 3 μL DNA, the LightCycler FastStart DNA Master SYBR Green I Kit (Roche Diagnostics, Mannheim, Germany) a LightCycler 2.0 real-time thermocycler, and a 10-fold serial dilution of parasite DNA to prepare a standard curve to allow quantification of parasite load, assuming 200 femtograms (fg) DNA as equivalent to 1 parasite [13,15]. Cycle threshold (Ct) of the positive samples was recorded.

### 2.6. Background Information on Reference and Index Tests

In a previous study, we determined the analytical sensitivity of these tests; LnPCR and qPCR could detect the equivalent of 10^−2^ parasites/μL, while Loopamp detected up to 10^−3^ parasites/μL [13]. The LnPCR and qPCR have been used for years at the WHO-CCL and by others for diagnosis of leishmaniasis and to support research and epidemiological studies [14,16,17,18,19,20,21,22].

### 2.7. Data Analysis

For sensitivity and specificity analyses, samples that had a positive result by LnPCR were considered cases, those with a negative result were considered controls. To determine the concordance and Cohen’s κ coefficient analysis, we defined true positive, true negative, false positive, and false negative samples among the different tests The correlation between qPCR Ct/parasite load and LAMP Tp was estimated using Pearson’s coefficient.

Sensitivity, specificity, and the concordance between tests (Cohen’s kappa coefficient), were calculated using Epidat v.3.0 [23] and GraphPad Prism v.7.0 software (GraphPad Software, San Diego, CA, USA). The correlations between the LAMP Tp, the qPCR Ct and parasite load, were determined using the same GraphPad Prism v.7.0 package.

### 2.8. Ethics

The WHO-CCL collection on leishmaniasis is registered at the National Biobank Register, Section Collections, Spain with the collection Reference ID: C.0000898. Ethical clearance to use these samples in evaluation of diagnostic tests is not required.

## 3. Results

The results of the repeat LnPCR were the same as when the samples were first tested, confirming the integrity of the samples selected for this study.

Details of the diagnostic performance of the tests used in this study are provided in Table 2. Loopamp results were identical when they were read by either real-time fluorimetry or direct inspection with the naked eye, both in CL and VL samples. The sensitivity of Loopamp for VL diagnosis was high, both on peripheral blood (97.4%) and bone marrow (97.9%) samples, as well as the specificity (>96%). This performance was quite similar to that of the qPCR, which reached higher sensitivity only on peripheral blood samples (100%), but with overlapping confidence intervals with Loopamp. In samples from CL suspected cases, Loopamp returned higher sensitivity than qPCR (100 vs. 98.2%), and both methods showed the highest specificity (100%).

When the results on samples from CL and VL suspected cases are taken together the three methods, Looamp, LnPCR and qPCR showed a very high concordance, as shown in Table 3.

When positive samples were analyzed, a strong correlation was observed between Loopamp time-to-positivity (Tp) and qPCR cycle threshold (Ct), r = 0.799 (95% CI 0.728–0.854), *p* < 0.0001 (Figure 1A). In addition, we also observed an excellent concordance between these two variables, C: 97.9% (95% CI 94.4–99.8); K: 0.94 (95% CI: 0.89–0.99). Likewise, there was a strong inverse correlation between Loopamp Tp and parasite load determined by qPCR, r = −0.673; 95% CI (−0.759 to −0.562); *p* < 0.0001 (Figure 1).

## 4. Discussion

The advantages of molecular tests to confirm *Leishmania* infection in CL and VL are well known, despite this its wider implementation has not been achieved. In Europe, where *L. infantum* causes CL and VL, PCR-based diagnosis is more frequent than in other regions affected by leishmaniasis, but its use is still mainly restricted to reference hospitals and research centers [24]. Standardized, robust and highly accurate tests like Loopamp™ *Leishmania* Detection Kit would facilitate the broad adoption of molecular diagnosis for leishmaniasis. We have previously shown that Loopamp allowed highly sensitive detection of *L. infantum* DNA using different methods for reading the results [13]. In this work we add data on clinical performance and show that highly sensitive diagnosis of CL (100%) and VL (>97%) is possible, with high specificity (>96% for VL and 100% for CL).

Our results add to previous studies using the same test, like Loopamp™ *Leishmania* Detection Kit. These showed high diagnostic accuracy for VL diagnosis due to *L. donovani* in Sudan (sensitivity: 100%; specificity: 99.01%) and Ethiopia (sensitivity: 92%; specificity: 100%) or CL due to *L. tropica* in Afghanistan (sensitivity: 92.2%; specificity: 94.1%), *L. guyanensis* in Suriname (sensitivity: 91.4%; specificity: 91.7%) and *L. panamiensis* in Colombia (sensitivity: 95%; specificity: 86%) [9,10,11,12].

Similarly, other LAMP tests have shown good performance in the diagnosis of human leishmaniasis, these targeting the same or different regions in the genome of *Leishmania*, being either species-specific or genus-specific, and using different methods for detection of amplified products. Yet, and to the best of our knowledge, the only LAMP test available as ready-to-use kit for the diagnosis of leishmaniasis, using stable dried reagents is the Loopamp™ *Leishmania* Detection Kit [25].

In the panel of clinical samples studied, Loopamp shows a diagnostic performance comparable to that of LnPCR and qPCR, this finding is not surprising as its limit of detection is equivalent to 10^−3^ parasites/μL (10^−2^ for LnPCR and qPCR) [13]. Another important aspect that we confirm is the strong correlation of Loopamp Tp with qPCR Ct. An earlier pilot study showed an excellent correlation between Loopamp Tp and qPCR Ct, and the Ct-generated parasite load, using a panel of promastigote-derived DNA samples and 10 clinical samples [13]. The present work confirms this with a larger sample of clinical specimens (*n* = 230). This opens the possibility of using Loopamp for additional applications in the management of leishmaniasis, such as monitoring parasite load to assess treatment efficacy or a pharmacodynamics tool in clinical trials.

## 5. Conclusions

The Loopamp™ *Leishmania* Detection Kit run on the Genie III real-time fluorimeter has a diagnostic performance comparable to that of two PCR methods used for routine diagnosis at the WHO-CCL and elsewhere, which provides support to the national health system in Spain for the diagnosis of leishmaniasis, as well as technical assistance to WHO and partners [26]. The Loopamp test is an accurate, simple, robust, standardized and CE-marked test; these are characteristics that should facilitate the adoption and implementation of molecular diagnosis for leishmaniasis.

## Figures and Tables

**Figure 1 microorganisms-09-00610-f001:**
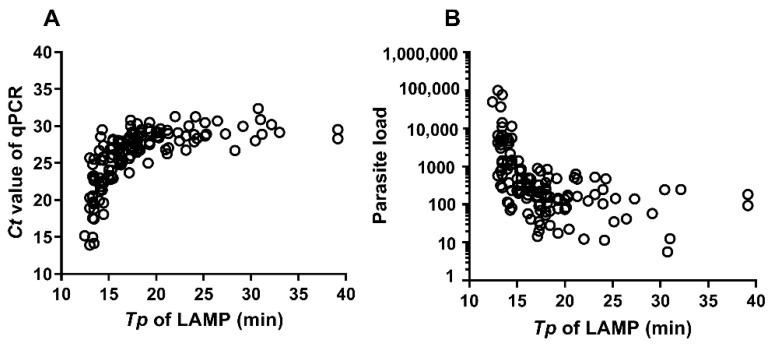
(**A**)Correlation (Spearman’s rank) between Loopamp Tp and qPCR Ct values. (**B**) correlation (inverse) between Loopamp Tp and parasite load. Ct: cycle threshold; qPCR: quantitative PCR; Tp: time-to-positivity; LAMP: Loopamp.

**Table 1 microorganisms-09-00610-t001:** Panel of samples used in this study, by clinical form (VL: visceral leishmaniasis. CL: cutaneous leishmaniasis) and result of the reference test (*Leishmania* nested PCR).

		Reference Test	
Condition	Sample Type	LnPCR Positive	LnPCR Negative
VL suspect	Peripheral blood	38	25
	Bone marrow	48	19
CL suspect	Skin biopsy	56	44

**Table 2 microorganisms-09-00610-t002:** Diagnostic Performance of Loopamp and qPCR on Samples from CL and VL Suspected Cases, Using LnPCR as Reference.

	Tests	LnPCR-Positive (Cases)	LnPCR-Negative (Controls)	Sensitivity (%) (95% CI)	Specificity (%) (95% CI)
VL (Peripheral Blood)	Loopamp-Positive	37	1	97.4 (0.9–100)	96 (86.3–100)
	Loopamp-Negative	1	24		
	qPCR-Positive	38	1	100 (98.7–100)	96 (86.3–100)
	qPCR-Negative	0	24		
VL (Bone Marrow)	Loopamp-Positive	47	0	97.9 (92.8–100)	100 (97.4–100)
	Loopamp-Negative	1	19		
	qPCR-Positive	47	0	97.9 (92.8–100)	100 (97.4–100)
	qPCR-Negative	1	19		
CL (Skin Biopsy)	Loopamp-Positive	56	0	100 (99.1–100)	100 (98.8–100)
	Loopamp-Negative	0	44		
	qPCR-Positive	55	0	98.2 (93.8–100)	100 (98.8–100)
	qPCR-Negative	1	44		

**Table 3 microorganisms-09-00610-t003:** Concordance of the Three Molecular Diagnostic Tests on Samples from CL and VL Suspected Cases.

	qPCR	LnPCR
**LnPCR**	C: 98.4% (94.4–99.8) K: 0.96 (0.91–1)	
**Loopamp**	C: 97.9% (93.2–99.5) K: 0.95 (0.9–1)	C: 98.4% (4.4–99.8) K: 0.96 (0.91–1)

C: concordance correlation coefficient; K value of Cohen’s kappa; 95% confidence intervals in brackets.

## Data Availability

The datasets used and analyzed during the current study are available in the manuscript or as Appendix A. Other data may be available from the corresponding author on reasonable request.

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
