# Peer review of "Loop-Mediated Isothermal Amplification Allows Rapid, Simple and Accurate Molecular Diagnosis of Human Cutaneous and Visceral Leishmaniasis Caused by Leishmania infantum When Compared to PCR"

_microorganisms, 2021, doi:10.3390/microorganisms9030610_

Round 1

Reviewer 1 Report

Interesting study evaluating the performance a new method for the identification of the parasite Leishmania in the biological tissues of patients with clinical disease. The paper is interesting and well written and, in the opinion of this reviewer, requires only some minor modifications.

Title: This is a comparison of methods. Please add in the title that the method was compared with real time PCR

Line 53: It is known that blood is a sample with reduced sensitivity compared to marrow or lymph nodes. Please emphasize this aspect. Furthermore, PCR does not lack standard protocols, rather it is a method that requires specialized laboratories and therefore little extensively usable.

Line 127: Statistical processing is unclear and should be described in more detail. You have used the PCR as reference method. It should be indicated that the two methods tested are compared with a reference method and their names should be indicated. This was used to define true positive and negative and to assess the sensitivity and specificity of test methods. Then the concordance between the two test methods was evaluated with the K test. All this must be explained in detail. The correlation with the Spearman test must also be reported in this paragraph.

Author Response

MANUSCRIPT 1143587 – Microorganisms

RESPONSE TO ACADEMIC EDITOR AND REVIEWERS

To the Editor and Reviewers,

We do appreciate that the journal Microorganisms has considered our work as a potential contribution to the special issues Leishmania and Leishmaniasis. And we also appreciate the constructive comments as well as the quick turnaround time.

We have addressed the points raised in the review and changed the manuscript after assessment of the same. When changes have incorporated, these are indicated in the manuscript or associated files using track changes or highlighting type colour. We also provide below a point-by-point response to the comments received. We hope we have interpreted and addresses your comments properly.

Should it be needed, we will be happy to clarify any of the points.

Sincerely,

Israel Cruz

Academic Editor note: "Dear Editors. If the paper is accepted to be published at microorganisms, be sure to tell the authors to correct supplementary

fig.1: skin biopsies come from patients  with CL suspected (not VL, this is a mistake at second level rectangle).

Also, correct the last negative (instead of nehtaiveat the last level rectangles)."

Figure 1 has been revised and typos addressed. Changes have been highlighted in red, please note this during edition and revert to black color. Applies to Skin biopsy 2nd level and last rectangles.

References section

By reviewing References section it seemed reference numbers were duplicated. We have edited this by removing duplicates. Lines 244-315.

Reviewer 1

Title: This is a comparison of methods. Please add in the title that the method was compared with real time PCR.

The title has been modified following the suggestion made by the reviewer. Lines 2-5.

Line 53: It is known that blood is a sample with reduced sensitivity compared to marrow or lymph nodes. Please emphasize this aspect. Furthermore, PCR does not lack standard protocols, rather it is a method that requires specialized laboratories and therefore little extensively usable.

We agree with the reviewer that traditionally the analysis of blood returns lower sensitivity than that obtained with more invasive procedures such as bone marrow, spleen or lymph node aspirates, mainly when parasitological tests such as microscopy and culture are used. But as it was highlighted in the systematic review in reference 7, the use of PCR has allowed circumventing this problem.

Nevertheless, we agree that this should be highlighted, as well as the use of PCR mainly in specialised laboratories. Following this suggestion we have modified the paragraph in lines 53-59 as follows:

“Molecular diagnosis based on polymerase chain reaction (PCR), has been widely used in the diagnosis of CL and VL and presents a number of advantages, such as increased sensitivity and the use of less invasive samples, showing high diagnostic accuracy for VL even using peripheral blood, a sample that usually yields lower sensitivity. Unfortunately, its cost and expertise required alongside the need for a cold chain and specialized laboratories, has prevented its implementation in most endemic settings [6,7].  “

Line 127: Statistical processing is unclear and should be described in more detail. You have used the PCR as reference method. It should be indicated that the two methods tested are compared with a reference method and their names should be indicated. This was used to define true positive and negative and to assess the sensitivity and specificity of test methods. Then the concordance between the two test methods was evaluated with the K test. All this must be explained in detail. The correlation with the Spearman test must also be reported in this paragraph.

We indicate that the reference test is LnPCR, described in section 2.3, starting on line 100. We also indicate that the index tests are Loopamp (described in section 2.4, starting on line 109) and qPCR (described in section 2.4, starting on line 119). In Table 1 we indicate that positives for LnPCR (reference test) are considered cases, and negatives as controls. But we acknowledge this information can be further detailed, therefore we have added more information and modified section 2.7 as follows, lines 133-144:

For sensitivity and specificity analyses, samples that had a positive result by LnPCR were considered cases, those with a negative result were considered controls. To determine the concordance and Cohen's κ coefficient analysis, we defined true pos-itive, true negative, false positive and false negative samples among the different tests The correlation between qPCR Ct/parasite load and LAMP Tp was estimated using Pearson's coefficient.

Sensitivity, specificity, and the concordance between tests (Cohen’s kappa coeffi-cient), were calculated using Epidat v.3.0 [23] and GraphPad Prism v.7.0 software (GraphPad Software, San Diego, CA, USA). The correlations between the LAMP Tp, the qPCR Ct and parasite load, were determined using the same GraphPad Prism v.7.0 package.

Reviewer 2

I suggest to talk more about Leishmaniasis, particularly cutaneous leishmaniasis and visceral leishmaniasis and to talk about their treatment, therefore I suggest to use this two references:

1-    Treatment of Visceral Leishmaniasis E M Moore and D N Lockwood J Glob Infect Dis. 2010 May-Aug; 2(2): 151–158. doi: 10.4103/0974-777X.62883

2-    Cutaneous Leishmaniasis: Current Treatment Practices in the USA for Returning Travelers Daniel P. Eiras, MD, MPH, Laura A. Kirkman, MD, and Henry W. Murray, MD Curr Treat Options Infect Dis. 2015 Mar 1; 7(1): 52–62. doi: 10.1007/s40506-015-0038-4

We do appreciate the suggestion, but we would like to highlight that the manuscript we are presenting is quite focused on the diagnostic application of molecular tests. This is why we have not entered in general aspects related to epidemiology and control, which, although important, we believe will distract the focus.

References: The authors did not follow perfectly the journal guidelines. (Abbreviated Journal Name Year, Volume, page range)

We do believe we have followed the guidelines of the journal, and when available (as also suggested by the journal) we have included the doi of the reference. Example:

Cruz I, Millet A, Carrillo E, Chenik M, Salotra P, Verma S, Veland N, Jara M, Adaui V, Castrillón C, Arévalo J, Moreno J, Cañavate C. An approach for interlaboratory comparison of conventional and real-time PCR assays for diagnosis of human leishmaniasis. Exp Parasitol. 2013 Jul;134(3):281-9. doi: 10.1016/j.exppara.2013.03.026

Reviewer 2 Report

This article aims to compare the diagnostic performance of Loopamp Leishmania Detection Kit (Eiken Chemical Co., Ltd., Japan) with conventional and real-time polymerase chain reaction (PCR) for human cutaneous and visceral leishmaniasis caused by L. infantum and to clarify that

 The performance of Loopamp is comparable to that of LnPCR and qPCR in the diagnosis of cutaneous and visceral leishmaniasis due to L. infantum.

 The manuscript is written comprehensively enough to be understandable.

The article stated the purpose, discussion and global implication are clearly stated and consistent with the rest of the manuscript.

The authors clearly described leishmaniasis, particularly cutaneous and visceral leishmaniasis in their introduction, and they addressed their hypothesis and opinion in a reproducible way, the study was presented in a clear way which facilitate in reaching accurate conclusions.

They present and interpret the literature and conclusions in an appropriate and comprehensive manner.

The conclusion summarizes all of the main points they stated throughout this review, no plagiarism has been noticed.

I suggest to talk more about

 Leishmaniasis, particularly cutaneous leishmaniasis and visceral leishmaniasis and to talk about their treatment, therefore I suggest to use this two references:

1-    Treatment of Visceral Leishmaniasis

E M Moore and D N Lockwood

J Glob Infect Dis. 2010 May-Aug; 2(2): 151–158.

doi: 10.4103/0974-777X.62883

2-    Cutaneous Leishmaniasis: Current Treatment Practices in the USA for Returning Travelers

Daniel P. Eiras, MD, MPH, Laura A. Kirkman, MD, and Henry W. Murray, MD

Curr Treat Options Infect Dis. 2015 Mar 1; 7(1): 52–62.

doi: 10.1007/s40506-015-0038-4

References: The authors did not follow perfectly the journal guidelines.

   (Abbreviated Journal Name YearVolume, page range)

Author Response

(The authors gave the same response as above.)
